# IDEAL: Inexact DEcentralized Accelerated Augmented Lagrangian Method

**Yossi Arjevani**
NYU
yossia@nyu.edu

**Joan Bruna**
NYU
bruna@cims.nyu.edu

**Bugra Can**
Rutgers University
bc600@scarletmail.rutgers.edu

**Mert Gürbüzbalaban**
Rutgers University
mg1366@rutgers.edu

**Stefanie Jegelka**
MIT
stefje@csail.mit.edu

**Hongzhou Lin**
MIT
hongzhou@mit.edu

## Abstract

We introduce a framework for designing primal methods under the decentralized optimization setting where local functions are smooth and strongly convex. Our approach consists of approximately solving a sequence of sub-problems induced by the accelerated augmented Lagrangian method, thereby providing a systematic way for deriving several well-known decentralized algorithms including EXTRA [47] and SSDA [43]. When coupled with accelerated gradient descent, our framework yields a novel primal algorithm whose convergence rate is optimal and matched by recently derived lower bounds. We provide experimental results that demonstrate the effectiveness of the proposed algorithm on highly ill-conditioned problems.

## 1 Introduction

Due to their rapidly increasing size, modern datasets are typically collected, stored and manipulated in a distributed manner. This, together with strict privacy requirements, has created a large demand for efficient solvers for the decentralized setting in which models are trained locally at each agent, and only local parameter vectors are shared. This approach has become particularly appealing for applications such as edge computing [30, 48], cooperative multi-agent learning [8, 38] and federated learning [31, 49]. Clearly, the nature of the decentralized setting prevents a global synchronization, as only communication within the neighboring machines is allowed. The goal is then to arrive at a consensus on all local agents with a model that performs as well as in the centralized setting.

Arguably, the simplest approach for addressing decentralized settings is to adapt the vanilla gradient descent method to the underlying network architecture [11, 18, 34, 53]. To this end, the connections between the agents are modeled through a mixing matrix which dictates how agents average over their neighbors' parameter vectors. Perhaps surprisingly, when the stepsizes are constant, simply averaging over the local iterates via the mixing matrix only converges to a neighborhood of the optimum [47, 58]. A recent line of works [17, 35, 36, 39, 46, 47] proposed a number of alternative methods that linearly converge to the global minimum. Extensions to the composite setting are also studied [1, 2, 27, 55].

The overall complexity of solving decentralized optimization problems is typically determined by two factors: (i) the condition number of the objective function $\kappa_f$, which measures the hardness of solving the underlying optimization problem, and (ii) the condition number of the mixing matrix $\kappa_W$, which quantifies the severity of 'information bottlenecks' present in the network. Lower complexity bounds recently derived for distributed settings [3, 5, 43, 52] show that one cannot expect to have a better dependence on the condition numbers than $\sqrt{\kappa_f}$ and $\sqrt{\kappa_W}$. Notably, despite the recent

considerable progress, none of the methods mentioned above is able to achieve accelerated rates, that is, a square root dependence for both $\kappa_f$ and $\kappa_W$—simultaneously.

An extensive effort has been devoted to obtaining acceleration for decentralized algorithms under various settings [12, 13, 16, 26, 40, 43, 44, 51, 54, 57, 59]. When a dual oracle is available, that is, access to the gradients of the dual functions is provided, optimal rates can be attained for smooth and strongly convex objectives [43]. However, having access to a dual oracle is a very restrictive assumption, and resorting to a direct 'primalization' through inexact approximation of the dual gradients leads to sub-optimal worst-case theoretical rates [51]. In this work, we propose a novel primal approach that leads to optimal rates in terms of dependency on $\kappa_f$ and $\kappa_W$.

Our contributions can be summarized as follows.

- We introduce a novel framework based on the accelerated augmented Lagrangian method for designing primal decentralized methods. The framework provides a simple and systematic way for deriving several well-known decentralized algorithms [17, 46, 47], including EXTRA [47] and SSDA [43], and unifies their convergence analyses.

- Using accelerated gradient descent as a sub-routine, we derive a novel method for smooth and strongly convex local functions which achieves optimal *accelerated* rates on both the condition numbers of the problem, $\kappa_f$ and $\kappa_W$, using *primal* updates, see Table 2.

- We perform a large number of experiments which confirm our theoretical findings, and demonstrate a significant improvement when the objective function is ill-conditioned and $\kappa_f \gg \kappa_W$.

## 2    Decentralized Optimization Setting

We consider $n$ computational agents and a network graph $\mathcal{G} = (\mathcal{V}, \mathcal{E})$ which defines how the agents are linked. The set of vertices $\mathcal{V} = \{1, \cdots, n\}$ represents the agents and the set of edges $\mathcal{E} \in \mathcal{V} \times \mathcal{V}$ specifies the connectivity in the network, i.e., a communication link between agents $i$ and $j$ exists if and only if $(i, j) \in \mathcal{E}$. Each agent has access to local information encoded by a loss function $f_i : \mathbb{R}^d \to \mathbb{R}$. The goal is to minimize the global objective over the entire network,

$$\min_{x \in \mathbb{R}^d} f(x) := \sum_{i=1}^n f_i(x). \tag{1}$$

In this paper, we assume that the local loss functions $f_i$ are differentiable, $L$-smooth and $\mu$-strongly convex.[1] Strong convexity of the component functions $f_i$ implies that the problem admits a unique solution, which we denote by $x^*$.

We consider the following computation and communication models [43]:

- **Local computation:** Each agent is able to compute the gradients of $f_i$ and the cost of this computation is one unit of time.

- **Communication:** Communication is done synchronously, and each agent can only exchange information with its neighbors, where $i$ is a neighbor of $j$ if $(i, j) \in \mathcal{E}$. The ratio between the communication cost and computation cost per round is denoted by $\tau$.

We further assume that propagation of information is governed by a mixing matrix $W \in \mathbb{R}^{n \times n}$ [34, 43, 58]. Specifically, given a local copy of the decision variable $x_i \in \mathbb{R}^d$ at node $i \in [1, n]$, one round of communication provides the following update $x_i \leftarrow \sum_{i=1}^n W_{ij} x_j$. The following standard assumptions regarding the mixing matrix [43] are made throughout the paper.

**Assumption 1.** *The mixing matrix $W$ satisfies the following:*

1. **Symmetry:** $W = W^T$.

2. **Positiveness:** $W$ *is positive semi-definite.*

3. **Decentralized property**: *If $(i, j) \notin \mathcal{E}$ and $i \neq j$, then $W_{ij} = W_{ji} = 0$.*

4. **Spectrum property:** *The kernel of $W$ is given by the vector of all ones $Ker(W) = \mathbb{R}\mathbf{1_n}$.*

**Algorithm 1** Decentralized Augmented Lagrangian framework

---
**Input:** mixing matrix $W$, regularization parameter $\rho$, stepsize $\eta$.
1: **for** $k = 1, 2, ..., K$ **do**
2:     $\mathbf{X}_k = \arg\min \left\{ P_k(\mathbf{X}) := F(\mathbf{X}) + \boldsymbol{\Lambda}_k^T \mathbf{X} + \frac{\rho}{2} \|\mathbf{X}\|_{\mathbf{W}}^2 \right\}$.
3:     $\boldsymbol{\Lambda}_{k+1} = \boldsymbol{\Lambda}_k + \eta \, \mathbf{W} \mathbf{X}_k$.
4: **end for**

---

A typical choice of the mixing matrix is the (weighted) Laplacian matrix of the graph. Another common choice is to set $W$ as $I - \tilde{W}$ where $\tilde{W}$ is a doubly stochastic matrix [7, 10, 47]. By Assumption 1.4, all the eigenvalues of $W$ are strictly positive, except for the smallest one. We let $\lambda_{\max}(W)$ denote the maximum eigenvalue, and let $\lambda_{\min}^+(W)$ denote the smallest positive eigenvalue. The ratio between these two quantities plays an important role in quantifying the overall complexity of this problem.

**Theorem 1** (Decentralized lower bound [43])**.** *For any first-order black-box decentralized method, the number of time units required to reach an $\epsilon$-optimal solution for (1) is lower bounded by*

$$\Omega \left( \sqrt{\kappa_f}(1 + \tau \sqrt{\kappa_W}) \log \left( \frac{1}{\epsilon} \right) \right), \tag{2}$$

*where $\kappa_f = L/\mu$ is the condition number of the loss function and $\kappa_W = \lambda_{\max}(W)/\lambda_{\min}^+(W)$ is the condition number of the mixing matrix.*

The lower bound decomposes as follows: a) **computation cost**, given by $\sqrt{\kappa_f} \log(1/\epsilon)$, and b) **communication cost**, given by $\tau \sqrt{\kappa_f \kappa_W} \log(1/\epsilon)$. The computation cost matches lower bounds for centralized settings [4, 37], while the communication cost introduces an additional term which depends on $\kappa_W$ and accounts for the 'price' of communication in decentralized models. It follows that the effective condition number of a given decentralized problem is $\kappa_W \kappa_f$.

Clearly, the choice of the matrix $W$ can strongly affect the optimal attainable performance. For example, $\kappa_W$ can get as large as $n^2$ in the line/cycle graph, or be constant in the complete graph. In this paper, we do not focus on optimizing over the choice of $W$ for a given graph $G$; instead, following the approach taken by existing decentralized algorithms, we assume that the graph $G$ and the mixing matrix W are given and aim to achieve the optimal complexity (2) for this particular choice of $W$.

## 3 Related Work and the Dual Formulation

A standard approach to address problem (1) is to express it as a constrained optimization problem

$$\min_{\mathbf{X} \in \mathbb{R}^{nd}} F(\mathbf{X}) := \frac{1}{n} \sum_{i=1}^{n} f_i(x_i) \quad \text{such that} \quad x_1 = x_2 = \cdots = x_n \in \mathbb{R}^d, \tag{P}$$

where $\mathbf{X} = [x_1; x_2; \cdots x_n] \in \mathbb{R}^{nd}$ is a concatenation of the vectors. To lighten the notation, we introduce the global mixing matrix $\mathbf{W} = W \otimes I_d \in \mathbb{R}^{nd \times nd}$, where $\otimes$ denotes the Kronecker product, and let $\|\cdot\|_{\mathbf{W}}$ denote the semi-norm induced by $\mathbf{W}$, i.e. $\|\mathbf{X}\|_{\mathbf{W}}^2 = \mathbf{X}^T \mathbf{W} \mathbf{X}$. With this notation in hand, we briefly review existing literature on decentralized algorithms.

**Decentralized Gradient Descent**    The decentralized gradient method [34, 58] has the update rule

$$\mathbf{X}_{k+1} = \mathbf{W} \mathbf{X}_k - \eta \nabla F(\mathbf{X}_k). \tag{DGD}$$

However, with constant stepsize, the algorithm does not converge to a global minimum of (P), but rather to a neighborhood of the solution [58]. A decreasing stepsize schedule may be used to ensure convergence but this yields a sublinear convergence rate, even in the strongly convex case.

**Linearly convergent primal algorithms**    By and large, recent methods that achieve linear convergence in the strongly convex case [17, 25, 35, 36, 39, 46, 47, 50] can be shown to follow a general framework based on the augmented Lagrangian method, see Algorithm 1. The main difference lies

---

**Algorithm 2** Accelerated Decentralized Augmented Lagrangian framework

---

**Input:** mixing matrix $W$, regularization parameter $\rho$, stepsize $\eta$, extrapolation parameters $\{\beta_k\}_{k \in \mathbb{N}}$
1: Initialize dual variables $\mathbf{\Lambda}_1 = \mathbf{\Omega}_1 = \mathbf{0} \in \mathbb{R}^{nd}$.
2: **for** $k = 1, 2, ..., K$ **do**
3: $\quad \mathbf{X}_k = \arg\min \left\{ P_k(\mathbf{X}) := F(\mathbf{X}) + \mathbf{\Omega}_k^T \mathbf{X} + \frac{\rho}{2} \|\mathbf{X}\|_{\mathbf{W}}^2 \right\}.$
4: $\quad \mathbf{\Lambda}_{k+1} = \mathbf{\Omega}_k + \eta \mathbf{W} \mathbf{X}_k$
5: $\quad \mathbf{\Omega}_{k+1} = \mathbf{\Lambda}_{k+1} + \beta_{k+1}(\mathbf{\Lambda}_{k+1} - \mathbf{\Lambda}_k)$
6: **end for**
**Output:** $\mathbf{X}_K$.

---

---

**Algorithm 3** IDEAL: Inexact Acc-Decentralized Augmented Lagrangian framework

---

**Additional Input:** A first-order optimization algorithm $\mathcal{A}$
Apply $\mathcal{A}$ to solve the subproblem $P_k$ warm starting at $\mathbf{X}_{k-1}$ to find an approximate solution

$$\mathbf{X}_k \approx \arg\min \left\{ P_k(\mathbf{X}) := F(\mathbf{X}) + \mathbf{\Omega}_k^T \mathbf{X} + \frac{\rho}{2} \|\mathbf{X}\|_{\mathbf{W}}^2 \right\},$$

**Option I:** stop the algorithm when $\|\mathbf{X}_k - \mathbf{X}_k^*\|^2 \leq \epsilon_k$, where $\mathbf{X}_k^*$ is the unique minimizer of $P_k$.
**Option II:** stop the algorithm after a prefixed number of iterations $T_k$.

---

in how subproblems $P_k$ are solved. Shi et al. [46] apply an alternating directions method; in [47], the EXTRA algorithm takes a single gradient descent step to solve $P_k$, see Appendix B for details. Jakovetić et al. [17] use multi-step algorithms such as Jacobi/Gauss-Seidel methods. To the best of our knowledge, the complexity of these algorithms is not better than $O\left((1+\tau)\kappa_f \kappa_W \log(\frac{1}{\epsilon})\right)$, in other words, they are non-accelerated. The recently proposed algorithm APM-C [26] enjoys a square root dependence on $\kappa_f$ and $\kappa_W$, but incurs an additional $\log(1/\epsilon)$ factor compared to the optimal attainable rate.

**Optimal method based on the dual formulation** By Assumption 1.4, the constraint $x_1 = x_2 = \cdots = x_n$ is equivalent to the identity $\mathbf{W} \cdot \mathbf{X} = 0$, which is again equivalent to $\sqrt{\mathbf{W}} \cdot \mathbf{X} = 0$. Hence, the dual formulation of (P) is given by

$$\max_{\mathbf{\Lambda} \in \mathbb{R}^{dn}} -F^*(-\sqrt{\mathbf{W}}\mathbf{\Lambda}). \tag{D}$$

Since the primal function is convex and the constraints are linear, we can use strong duality and address the dual problem instead of the primal one. Using this approach, [43] proposed a dual method with optimal accelerated rates, using Nesterov's accelerated gradient method for the dual problem (D). As mentioned earlier, the main drawback of this method is that it requires access to the gradient of the dual function which, unless the primal function has a relatively simple structure, is not available. One may apply a first-order method to approximate the dual gradients inexactly at the expense of an additional $\sqrt{\kappa_f}$ factor in the computation cost [51], but this woul make the algorithm no longer optimal. This indicates that achieving optimal rates when using primal updates is a rather challenging task in the decentralized setting. In the following sections, we provide a generic framework which allows us to derive a primal decentralized method with optimal complexity guarantees.

## 4 An Inexact Accelerated Augmented Lagrangian framework

In this section, we introduce our inexact accelerated Augmented Lagrangian framework, and show how to combine it with Nesterov's acceleration. To ease the presentation, we first describe a conceptual algorithm, Algorithm 2, where subproblems are solved exactly, and only then introduce inexact inner-solvers.

Similarly to Nesterov's accelerated gradient method, we use an extrapolation step for the dual variable $\mathbf{\Lambda}_k$. The component $\mathbf{W}\mathbf{X}_k$ in line 4 of Algorithm 2 is the negative gradient of the Moreau-envelope[2] of the dual function. Hence our algorithm is equivalent to applying Nesterov's method

on the Moreau-envelope of the dual function, or equivalently, an accelerated dual proximal point algorithm. This renders the optimal dual method proposed in [43] as a special case of our algorithmic framework (with $\rho$ set to 0).

While Algorithm 2 is conceptually plausible, it requires an exact solution of the Augmented Lagrangian problems, which can be too expensive in practice. To address this issue, we introduce an inexact version, shown in Algorithm 3, where the $k$-th subproblem $P_k$ is solved up to a predefined accuracy $\epsilon_k$. The choice of $\epsilon_k$ is rather subtle. On the one hand, choosing a large $\epsilon_k$ may result in a non-converging algorithm. On the other hand, choosing a small $\epsilon_k$ can be exceedingly expensive as the optimal solution of the subproblem $\mathbf{X}_k^*$ is not the global optimum $\mathbf{X}^*$. Intuitively, $\epsilon_k$ should be chosen to be of the same order of magnitude as $\|\mathbf{X}_k^* - \mathbf{X}^*\|$, leading to the following result.

**Theorem 2.** *Consider the sequence of primal variables $(\mathbf{X}_k)_{k \in \mathbb{N}}$ generated by Algorithm 3 with the subproblem $P_k$ solved up to $\epsilon_k$ accuracy in Option I. With parameters set to*

$$\beta_k = \frac{\sqrt{L_\rho} - \sqrt{\mu_\rho}}{\sqrt{L_\rho} + \sqrt{\mu_\rho}}, \quad \eta = \frac{1}{L_\rho}, \quad \epsilon_k = \frac{\mu_\rho}{2\lambda_{\max}(W)} \left( 1 - \frac{1}{2}\sqrt{\frac{\mu_\rho}{L_\rho}} \right)^k \Delta_{dual}, \tag{3}$$

*where $L_\rho = \frac{\lambda_{\max}(W)}{\mu + \rho\lambda_{\max}(W)}$, $\mu_\rho = \frac{\lambda_{\min}^+(W)}{L + \rho\lambda_{\min}^+(W)}$ and $\Delta_{dual}$ is the initial dual function gap, we obtain*

$$\|\mathbf{X}_k - \mathbf{X}^*\|^2 \le C_\rho \left( 1 - \frac{1}{2}\sqrt{\frac{\mu_\rho}{L_\rho}} \right)^k \Delta_{dual}, \tag{4}$$

*where $\mathbf{X}^* = \mathbf{1}_n \otimes x^*$ and $C_\rho = 258\frac{L_\rho\lambda_{\max}(W)}{\mu^2\mu_\rho^2}$.*

**Corollary 3.** *The number of subproblems $P_k$ to achieve $\|\mathbf{X}_k - \mathbf{X}^*\|^2 \le \epsilon$ in IDEAL is bounded by*

$$K = O\left( \sqrt{\frac{L_\rho}{\mu_\rho}} \log\left( \frac{C_\rho \Delta_{dual}}{\epsilon} \right) \right). \tag{5}$$

We remark that inexact accelerated Augmented Lagrangian methods have been previously analyzed under different assumptions [21, 33, 56]. The main difference is that here, we are able to establish a linear convergence rate, whereas existing analyses only yield sublinear rates. One of the reasons for this discrepancy is that, although $F^*$ is strongly convex, the dual problem (D) is not, as the mixing matrix $\mathbf{W}$ is singular. The key to obtaining a linear convergence rate is a fine-grained analysis of the dual problem, showing that the dual variables always lie in the subspace where strong convexity holds. The proof of the theorem relies on the equivalence between Augmented Lagrangian methods and the dual proximal point algorithm [9, 41], which can be interpreted as applying an inexact accelerated proximal point algorithm [15, 28] to the dual problem. A complete convergence analysis is deferred to Section C in the appendix.

Theorem 2 provides an accelerated convergence rate with respect to the 'augmented' condition number $\kappa_\rho := L_\rho/\mu_\rho$, as determined by the Augmented Lagrangian parameter $\rho$ in Algorithm 3. We have the following bounds:

$$\underbrace{1}_{\rho=\infty} \le \kappa_\rho = \frac{L + \rho\lambda_{\min}^+(W)}{\mu + \rho\lambda_{\max}(W)} \frac{\lambda_{\max}(W)}{\lambda_{\min}^+(W)} \le \underbrace{\frac{L}{\mu} \frac{\lambda_{\max}(W)}{\lambda_{\min}^+(W)}}_{\rho=0} = \kappa_f \kappa_W, \tag{6}$$

where we observe that the condition number $\kappa_\rho$ is a decreasing function of the regularization parameter $\rho$. When $\rho = 0$, the maximum value is attained at $\kappa_\rho = \kappa_f \kappa_W$, the effective condition number of the decentralized problem. As $\rho$ goes to infinity, the augmented condition number $\kappa_\rho$ goes to 1. Naively, one may want to take $\rho$ as large as possible to get a fast convergence. However, one must also take into account the complexity of solving the subproblems. Indeed, since $W$ is singular, the additional regularization term in $P_k$ does not improve the strong convexity of the subproblems, yielding an increase in inner loops complexity as $\rho$ grows. Hence, the optimal choice of $\rho$ requires balancing the inner and outer complexity in a careful manner.

To study the inner loop complexity, we introduce a warm-start strategy. Intuitively, the distance between $\mathbf{X}_{k-1}$ and the $k$-th solution $\mathbf{X}_k^*$ to the subproblem $P_k$ is roughly on the order of $\epsilon_{k-1}$. More precisely, we have the following result.

|  | GD | AGD | SGD |
|---|---|---|---|
| $T_k$ | $\tilde{O}\left(\frac{L+\rho\lambda_{\max}(W)}{\mu}\right)$ | $\tilde{O}\left(\sqrt{\frac{L+\rho\lambda_{\max}(W)}{\mu}}\right)$ | $\tilde{O}\left(\frac{\sigma^2}{\mu^2\epsilon_k}\right)$ |
| $\rho$ | $\frac{L}{\lambda_{\max}(W)}$ | $\frac{L}{\lambda_{\max}(W)}$ | $\frac{L}{\lambda_{\min}^+(W)}$ |
| $\sum_{k=1}^{K} T_k$ | $\tilde{O}\left(\kappa_f\sqrt{\kappa_W}\log(\frac{1}{\epsilon})\right)$ | $\tilde{O}\left(\sqrt{\kappa_f\kappa_W}\log(\frac{1}{\epsilon})\right)$ | $\tilde{O}\left(\frac{\sigma^2\kappa_f\kappa_W}{\mu^2\epsilon}\right)$ |

Table 1: The first row indicates the number of iterations required for different inner solvers to achieve $\epsilon_k$ accuracy for the $k$-th subproblem $P_k$; the $\tilde{O}$ notation hides logarithmic factors in the parameters $\rho$, $\kappa_f$ and $\kappa_W$. The second row shows the theoretical choice of the regularization parameter $\rho$. The last row shows the total number of iterations according to the choice of $\rho$.

**Lemma 4.** *Given the parameter choice in Theorem 2, initializing the subproblem $P_k$ at $\mathbf{X}_{k-1}$ yields,*

$$\|\mathbf{X}_{k-1} - \mathbf{X}_k^*\|^2 \leq \frac{8C_\rho}{\mu_\rho}\epsilon_{k-1}.$$

Consequently, the ratio between the initial gap at the $k$-th subproblem and the desired gap $\epsilon_k$ is bounded by

$$\frac{\|\mathbf{X}_{k-1} - \mathbf{X}_k^*\|^2}{\epsilon_k} \leq \frac{8C_\rho}{\mu_\rho}\frac{\epsilon_{k-1}}{\epsilon_k} \leq \frac{16C_\rho}{\mu_\rho} = O(\kappa_f\kappa_W\rho^2),$$

which is independent of $k$. In other words, the inner loop solver only needs to decrease the iterate gap by a constant factor for each $P_k$. If the algorithm enjoys a linear convergence rate, a constant number of iteration is sufficient for that. If the algorithm enjoys a sublinear convergence, then the inner loop complexity grows with $k$. To illustrate the behaviour of different algorithms, we present the inner loop complexity $T_k$ for gradient descent (GD), accelerated gradient descent (AGD) and stochastic gradient descent (SGD) in Table 1. Note that while the inner complexity of GD and AGD are independent of $k$, the inner complexity for SGD increases geometrically with $k$. Other possible choices for inner solvers are the alternating directions or Jacobi/Gauss-Seidel method, both of which yield accelerated variants for [46] and [17].

In fact, the theoretical upper bounds on the inner complexity also provide a more practical way to halt the inner optimization processes (see Option II in Algorithm 3). Indeed, one can predefine the computational budget for each subproblem, for instance, 100 iterations of AGD. If this budget exceeds the theoretical inner complexity $T_k$ in Table 1, then the desired accuracy $\epsilon_k$ is guaranteed to be reached. In particular, we do not need to evaluate the sub-optimality condition, it is automatically satisfied as long as the budget is chosen appropriately.

Finally, the global complexity is obtained by summing $\sum_{k=1}^{K} T_k$, where $K$ is the number of subproblems given in (5). Note that, so far, our analysis applies to any regularization parameter $\rho$. Since $\sum_{k=1}^{K} T_k$ is a function of $\rho$, this implies that one can select the parameter $\rho$ such that the overall complexity is minimized, leading to the choices of $\rho$ described in Table 1.

**Two-fold acceleration** In our setting, acceleration seems to occur in two stages (when compared to the non-accelerated $O\left(\kappa_f\kappa_W\log(\frac{1}{\epsilon})\right)$ rates in [17, 35, 39, 46, 47]). First, combining IDEAL with GD improves the dependence on the condition of the mixing matrix $\kappa_W$. Secondly, when used as an inner solver, AGD improves the dependence on the condition number of the local functions $\kappa_f$. This suggests that the two phenomena are independent; while one is related to the consensus between the agents, as governed by the mixing matrix, the other one is related to the respective centralized hardness of the optimization problem.

**Stochastic oracle** Our framework also subsumes the stochastic setting, where only noisy gradients are available. In this case, since SGD is sublinear, the required iteration counters $T_k$ for the subproblem must increase inversely proportional to $\epsilon_k$. Also the stepsize at the $k$-th iteration needs to be decreased accordingly. The overall complexity is now given by $\tilde{O}\left(\frac{\sigma^2\kappa_f\kappa_W}{\mu^2\epsilon}\right)$. However, in this case, the resulting dependence on the graph condition number can be improved [13].

|  | $\rho$ | Computation cost | Communication cost |
| --- | --- | --- | --- |
| SSDA+AGD | $0$ | $\tilde{O}\left(\kappa_f\sqrt{\kappa_W}\log(\frac{1}{\epsilon})\right)$ | $O\left(\tau\sqrt{\kappa_f\kappa_W}\log(\frac{1}{\epsilon})\right)$ |
| IDEAL+AGD | $\frac{L}{\lambda_{\max}(W)}$ | $\tilde{O}\left(\sqrt{\kappa_f\kappa_W}\right)\log(\frac{1}{\epsilon})$ | $\tilde{O}\left(\tau\sqrt{\kappa_f\kappa_W}\log(\frac{1}{\epsilon})\right)$ |
| MSDA+AGD | $0$ | $\tilde{O}\left(\kappa_f\log(\frac{1}{\epsilon})\right)$ | $O\left(\tau\sqrt{\kappa_f\kappa_W}\log(\frac{1}{\epsilon})\right)$ |
| MIDEAL+AGD | $\frac{L}{\lambda_{\max}(Q(W))}$ | $\tilde{O}\left(\sqrt{\kappa_f}\log(\frac{1}{\epsilon})\right)$ | $\tilde{O}\left(\tau\sqrt{\kappa_f\kappa_W}\log(\frac{1}{\epsilon})\right)$ |

Table 2: The communication cost of the presented algorithms are all optimal, but the computation cost differs. An additional factor of $\sqrt{\kappa_f}$ is introduced in SSDA/MSDA compared to their original rate in [43], due to the gradient approximation. The optimal computation cost is achieved by combining our multi-stage algorithm MIDEAL with AGD as an inner solver.

**Multi-stage variant (MIDEAL)**   We remark that the complexity presented in Table 2 is abbreviated, in the sense that it does not distinguish between communication cost and computation cost. To provide a more fine-grained analysis, it suffices to note that performing a gradient step of the subproblem $\nabla P_k(\mathbf{X}) = \nabla F(\mathbf{X}) + \mathbf{\Omega}_k + \rho\mathbf{W}\mathbf{X}$ requires one local computation to evaluate $\nabla F$, and one round of communication to obtain $\mathbf{W}\mathbf{X}$. This implies that when GD/AGD/SGD is combined with IDEAL, the number of local computation rounds is roughly the number of communication rounds, leading to a sub-optimal computation cost, as shown in Table 2.

To achieve optimal accelerated rates, we enforce multiple communication rounds after one evaluation of $\nabla F$. This is achieved by substituting the regularization metric $\|\cdot\|^2_{\mathbf{W}}$ with $\|\cdot\|^2_{Q(\mathbf{W})}$, where $Q$ is a well-chosen polynomial. In this case, the gradient of the subproblem becomes $\nabla P_k(\mathbf{X}) = \nabla F(\mathbf{X}) + \mathbf{\Omega}_k + \rho\,Q(\mathbf{W})\mathbf{X}$, which requires $\deg(Q)$ rounds of communication.

The choice of the polynomial $Q$ relies on Chebyshev acceleration, which is introduced in [6, 43]. More concretely, the Chebyshev polynomials are defined by the recursion relation $T_0(x) = 1$, $T_1(x) = x$, $T_{j+1}(x) = 2xT_j(x) - T_{j-1}(x)$, and $Q$ is defined by

$$Q(x) = 1 - \frac{T_{j_W}(c(1-x))}{T_{j_W}(c)} \quad \text{with} \quad j_W = \lfloor\sqrt{\kappa_W}\rfloor, \quad c = \frac{\kappa_W+1}{\kappa_W-1}. \tag{7}$$

Applying this specific choice of $Q$ to the mixing matrix $W$ reduces its condition number by the maximum amount [6, 43], yielding a graph independent bound $\kappa_{Q(W)} = \lambda_{\max}(Q(W))/\lambda^+_{\min}(Q(W)) \leq 4$. Moreover, the symmetry, positiveness and spectrum property in Assumption 1 are maintained by $Q(W)$. Even though $Q(W)$ no longer satisfies the decentralized property, it can be implemented using $\lfloor\sqrt{\kappa_W}\rfloor$ rounds of communications with respect to $W$. The implementation details of the resulting algorithm are similar to Algorithm 2, and follow by substituting the mixing matrix $W$ by $Q(W)$ (Algorithm 5 in Appendix E).

**Comparison with inexact SSDA/MSDA [43]**   Recall that SSDA/MSDA are special cases of our algorithmic framework with the degenerate regularization parameter $\rho = 0$. Therefore, our complexity analysis naturally extends to an inexact anlysis of SSDA/MSDA, as shown in Table 2. although the resulting communication costs are optimal, the computation cost is not, due to the additional $\sqrt{\kappa_f}$ factor introduced by solving the subproblems inexactly. In contrast, our multi-stage framework achieves the optimal computation cost.

- **Low communication cost regime:** $\tau\sqrt{\kappa_W} < 1$, the computation cost dominates the communication cost, a $\sqrt{\kappa_f}$ improvement is obtained by MIDEAL comparing to MSDA.

- **Ill conditioned regime:** $1 < \tau\sqrt{\kappa_W} < \sqrt{\kappa_f}$, the complexity of MSDA is dominated by the computation cost $\tilde{O}\left(\kappa_f\log(\frac{1}{\epsilon})\right)$ while the complexity MIDEAL is dominated by the communication cost $\tilde{O}\left(\tau\sqrt{\kappa_f\kappa_W}\log(\frac{1}{\epsilon})\right)$. The improvement is proportional to the ratio $\sqrt{\kappa_f}/\tau\sqrt{\kappa_W}$.

- **High communication cost regime:** $\sqrt{\kappa_f} < \tau\sqrt{\kappa_W}$, the communication cost dominates, and MIDEAL and MSDA are comparable.

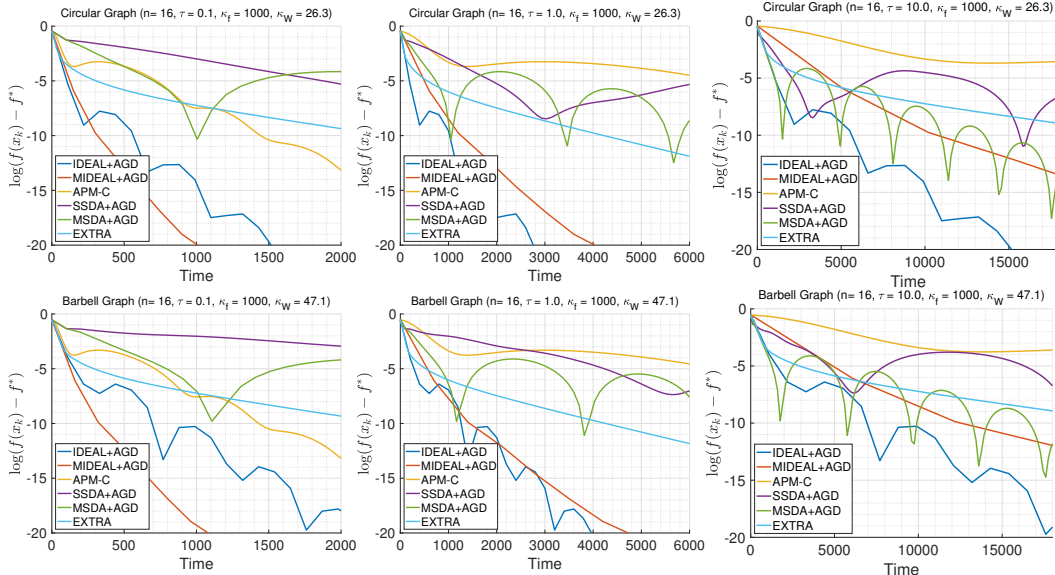

Figure 1: We evaluate the empirical performance of existing state-of-the-art algorithms, where the underlying network is a circular graph (top) and a barbell graph (bottom). We consider the following regimes: low communication cost (left), Ill-condition problems (middle) and High communication cost (right). The x-axis is the time counter, i.e. the sum of the communication cost and the computation cost; the y-axis is the log scale suboptimality. We observe that our algorithms IDEAL/MIDEAL are optimal under various regimes, validating our theoretical findings.

## 5    Experiments

Having described the IDEAL/MIDEAL algorithms for decentralized optimization problem (1), we now turn to presenting various empirical results which corroborate our theoretical analysis. To facilitate a simple comparison between existing state-of-the-art algorithms, we consider an $\ell_2$-regularized logistic regression task over two classes of the MNIST [24]/CIFAR-10 [23] benchmark datasets.

For the MNIST experiment, we directly use the normalized image as input feature. For the CIFAR experiment, a linear model is not rich enough to express the complex images. Hence, we first apply an unsupervised learning model, convolutional kernel network [29], to extract the feature and then apply the logistic regression on top of it. This can be approximately viewed as training the last layer of a conventional neural network by freezing the well-trained first layers.

The smoothness parameter of the logistic regression (assuming normalized feature vectors) can be shown to be bounded by $1/4$, which together with a regularization parameter $\mu \approx 1\mathrm{e}{-3}$, yields a relatively high $1\mathrm{e}3$-bound on the condition number of the loss function. Further empirical results which demonstrate the robustness of IDEAL/MIDEAL under wide range of parameter choices are provided in Appendix G.

We compare the performance of IDEAL/MIDEAL with the state-of-the-art algorithms EXTRA [47], APM-C [26] and the inexact dual method SSDA/MSDA [43]. We set the inner iteration counter to be $T_k = 100$ for all algorithms, and use the theoretical stepsize schedule. The decentralized environment is modelled in a synthetic setting, where the communication time is steady and no latency is encountered. To demonstrate the effect of the underlying network architecture, we consider: a) a circular graph, where the agents form a cycle; b) a Barbell graph, where the agents are split into two complete subgraphs, connected by a single bridge (shown in Figure 2 in the appendix).

As shown in Figure 1, our multi-stage algorithm MIDEAL is optimal in the regime where the communication cost $\tau$ is small, and the single-stage variant IDEAL is optimal when $\tau$ is large. As expected, the inexactness mechanism significantly slows down the dual method SSDA/MSDA in the low communication cost regime. In contrast, the APM-C algorithm performs reasonably well in the low communication regime, but performs relatively poorly when the communication cost is high.

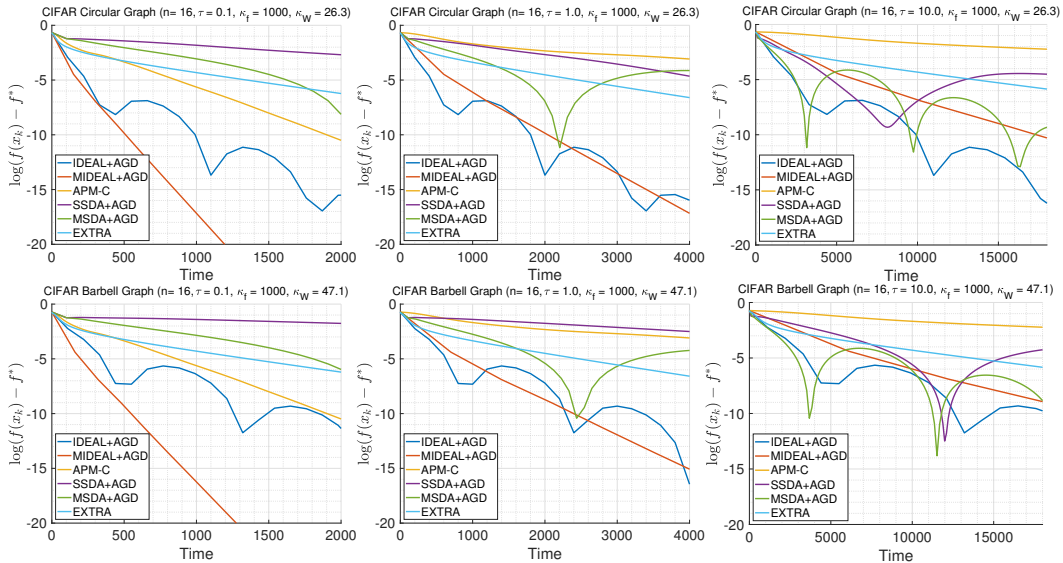

Figure 2: **CIFAR experiments**: We observe similar phenomenon as in the MNIST experiment, that the multi-stage algorithm MIDEAL outperforms when the communication cost $\tau$ is low and the IDEAL outperforms in the other cases.

## 6   Conclusions

We propose a novel framework of decentralized algorithms for smooth and strongly convex objectives. The framework provides a unified viewpoint of several well-known decentralized algorithms and, when instantiated with AGD, achieves optimal convergence rates in theory and state-of-the-art performance in practice. We leave further generalization to (non-strongly) convex and non-smooth objectives to future work.

## Acknowledgements

YA and JB acknowledge support from the Sloan Foundation and Samsung Research. BC and MG acknowledge support from the grants NSF DMS-1723085 and NSF CCF-1814888. HL and SJ acknowledge support by The Defense Advanced Research Projects Agency (grant number YFA17 N66001-17-1-4039). The views, opinions, and/or findings contained in this article are those of the author and should not be interpreted as representing the official views or policies, either expressed or implied, of the Defense Advanced Research Projects Agency or the Department of Defense.

## Broader impact

Centralization of data is not always possible because of security and legacy concerns [14]. Our work proposes a new optimization algorithm in the decentralized setting, which can learn a model without revealing the privacy sensitive data. Potential applications include data coming from healthcare, environment, safety, etc, such as personal medical information [19, 20], keyboard input history [22, 32] and beyond.

## Footnotes

[1] $f$ is $L$-smooth if $\nabla f$ is $L$-Lipschitz; $f$ is $\mu$-strongly convex if $f - \frac{\mu}{2}\|x\|^2$ is convex.

[2]A proper definition of the Moreau-envelope is given in [42], readers that are not familiar with this concept could take it as an implicit function which shares the same optimum as the original function.

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
