[Supplementary Material]

## A    Remark on the choice of the mixing matrix

In the main paper, the mixing matrix $W$ is defined following the convention used in [43], where the kernel of $W$ is the vector of all ones. It is worth noting that the term mixing matrix is also used in the literature to denote a doubly stochastic matrix $W_{DS}$ (see e.g. [10, 17, 35, 36, 39, 46, 47]). These two approaches are equivalent as given a doubly stochastic matrix $W_{DS}$, the matrix

$$I - W_{DS} \text{ is a mixing matrix under Definition 1.}$$

In the following discussion, we will use $W_{DS}$ to draw the connection when necessary.

## B    Recovering EXTRA under the augmented Lagrangian framework

The goal of this section is to show that EXTRA algorithm [47] is a special case of the non-accelerated Augmented Lagrangian framework in Algorithm 1.

**Proposition 5.** *The EXTRA algorithm is equivalent to applying one step of gradient descent to solve the subproblem in Algorithm 1.*

*Proof.* Taking a single step of gradient descent in the subproblem $P_k$ in Algorithm 1 warm starting at $X_{k-1}$ yields the update

$$X_k = X_{k-1} - \alpha(\nabla F(X_{k-1}) + \Lambda_k + \rho W X_{k-1}). \tag{8}$$
$$\Lambda_{k+1} = \Lambda_k + \eta W X_k.$$

Using the $(k+1)$-th update,

$$X_{k+1} = X_k - \alpha(\nabla F(X_k) + \Lambda_{k+1} + \rho W X_k). \tag{9}$$

and subtracting (8) from (9) gives

$$X_{k+1} = (2 - \alpha(\rho + \eta)W)X_k - (1 - \alpha\rho W)X_{k-1} - \alpha(\nabla F(X_k) - \nabla F(X_{k-1})).$$

When incorporating with the mixing matrix $W = I - W_{DS}$ and taking $\rho = \eta = \frac{1}{2\alpha}$ gives,

$$X_{k+1} = (I + W_{DS})X_k - \left(I + \frac{W_{DS}}{2}\right)X_{k-1} - \alpha(\nabla F(X_k) - \nabla F(X_{k-1})),$$

which is the update rule of EXTRA [47]. $\square$

**Remark 6.** *When expressing the parameters in terms of $\rho$, the inner loop stepsize reads as $\alpha = \frac{1}{2\rho}$, and the outer-loop stepsize reads as $\eta = \rho$.*

## C    Proof of Theorem 3

---

**Algorithm 4** (Unscaled) Accelerated Decentralized Augmented Lagrangian framework

---

**Input:** mixing matrix $W$, regularization parameter $\rho$, stepsize $\eta$, extrapolation parameters $\{\beta_k\}_{k \in \mathbb{N}}$
1: Initialize dual variables $\mathbf{\Lambda}_1 = \mathbf{\Omega}_1 = \mathbf{0} \in \mathbb{R}^{nd}$.
2: **for** $k = 1, 2, ..., K$ **do**
3:     $\mathbf{X}_k \approx \arg\min \left\{ P_k(\mathbf{X}) := F(\mathbf{X}) + (\sqrt{\mathbf{W}}\mathbf{\Omega}_k)^T \mathbf{X} + \frac{\rho}{2}\|\mathbf{X}\|_{\mathbf{W}}^2 \right\}$.
4:     $\mathbf{\Lambda}_{k+1} = \mathbf{\Omega}_k + \eta\sqrt{\mathbf{W}}\mathbf{X}_k$
5:     $\mathbf{\Omega}_{k+1} = \mathbf{\Lambda}_{k+1} + \beta_{k+1}(\mathbf{\Lambda}_{k+1} - \mathbf{\Lambda}_k)$
6: **end for**
**Output:** $\mathbf{X}_K$.

---

We start by noting that Algorithm 2 is equivalent to the "unscaled" version of Algorithm 4. More specifically, we recover Algorithm 2 by substituting the variables

$$\Lambda \leftarrow \sqrt{\mathbf{W}}\Lambda, \quad \Omega \leftarrow \sqrt{\mathbf{W}}\Omega.$$

The unscaled version is computationally inefficient since it requires the computation of the square root of $W$. This is the reason why we choose to present the scaled version Algorithm 2 in the main paper. However, the unscaled version is easier to work with for the analysis. **In the following proof, the variables $\Lambda$ and $\Omega$ are referred to as in the unscaled version Algorithm 4.**

The key concept underlying our analysis on is the Moreau-envelope of the dual problem:

$$\Phi_\rho(\Lambda) = \min_{\Gamma \in \mathbb{R}^{nd}} \left\{ F^*(-\sqrt{W}\Gamma) + \frac{1}{2\rho}\|\Gamma - \Lambda\|^2 \right\}. \tag{10}$$

Similarly, we define the associated proximal operator

$$\text{prox}_{\Phi_\rho}(\Lambda) = \arg\min_{\Gamma \in \mathbb{R}^{nd}} \left\{ F^*(-\sqrt{W}\Gamma) + \frac{1}{2\rho}\|\Gamma - \Lambda\|^2 \right\}. \tag{11}$$

Note that when the inner problem is strongly convex, the proximal operator is unique (that is, a single-valued operator). The following is a list well known properties of the Moreau-envelope:

**Proposition 7.** *The Moreau envelope $\Phi_\rho$ enjoys the following properties*

1. $\Phi_\rho$ *is convex and it shares the same optimum as the dual problem (D).*

2. $\Phi_\rho$ *is differentiable and the gradient of $\Phi_\rho$ is given by*

$$\nabla\Phi_\rho(\mathbf{\Lambda}) = \frac{1}{\rho}(\mathbf{\Lambda} - \text{prox}_{\Phi_\rho}(\mathbf{\Lambda}))$$

3. *If $F$ is twice differentiable, then its convex conjugate $F^*$ is also twice differentiable. In this case, $\Phi_\rho$ is also twice differentiable and the Hessian is given by*

$$\nabla^2\Phi_\rho(\Lambda) = \frac{1}{\rho}I - \frac{1}{\rho^2}\left[\frac{1}{\rho}I + \sqrt{\mathbf{W}}\nabla^2 F^*(-\sqrt{\mathbf{W}}\,\text{prox}_{\Phi_\rho}(\Lambda))\sqrt{\mathbf{W}}\right]^{-1}.$$

**Corollary 8.** *The Moreau envelope $\Phi_\rho$ satisfies*

1. $\Phi_\rho$ *is $L_\rho$-smooth, where $L_\rho = \frac{\lambda_{\max}(W)}{\mu + \rho\lambda_{\max}(W)} \leq \frac{1}{\rho}$.*

2. $\Phi_\rho$ *is $\mu_\rho$-strongly convex in the image space of $\sqrt{W}$, where $\mu_\rho = \frac{\lambda_{\min}^+(W)}{L + \rho\lambda_{\min}^+(W)}$.*

*Proof.* These properties follow from the expressions for the Hessian of $\Phi_\rho$ and by the fact that $F^*$ is $\frac{1}{\mu}$-smooth and $\frac{1}{L}$ strongly convex. $\qquad\square$

In particular, $\Phi_\rho$ is only strongly convex on the image space of $\sqrt{W}$, one of the keys to prove the linear convergence rate is the following lemma.

**Lemma 9.** *The variables $\mathbf{\Lambda}_k$ and $\mathbf{\Omega}_k$ in the un-scaled version Algorithm 4 all lie in the image space of $\sqrt{W}$ for any $k$.*

*Proof.* This can be easily derived by induction according to the update rule in line 4, 5 of Algorithm 4. $\qquad\square$

Similar to the dual Moreau-envelope, we also define the weighted Moreau-envelope on the primal function

$$\Psi_\rho(\Omega) = \min_{\mathbf{X}} \left\{ F(\mathbf{X}) + \Omega^T\mathbf{X} + \frac{\rho}{2}\|\mathbf{X}\|_{\mathbf{W}}^2 \right\} \tag{12}$$

and its associated proximal operator

$$\text{prox}_{\Psi_\rho}(\Omega) = \arg\min_{\mathbf{X}} \left\{ F(\mathbf{X}) + \Omega^T\mathbf{X} + \frac{\rho}{2}\|\mathbf{X}\|_{\mathbf{W}}^2 \right\}. \tag{13}$$

Indeed, this function corresponds exactly to the subproblem solved in the augmented Lagrangian framework (line 3 of Algorithm 2). Similar property holds for $\Psi_\rho$:

**Proposition 10.** *The Moreau envelope* $\Psi_\rho$ *enjoys the following properties:*

1. $\Psi_\rho$ *is concave.*

2. $\Psi_\rho$ *is differentiable and the gradient of* $\Psi_\rho$ *is given by*

$$\nabla\Psi_\rho(\Omega) = \text{prox}_\Psi(\Omega).$$

3. *If* $F$ *is twice differentiable, then* $\Psi_\rho$ *is also twice differentiable and the Hessian is given by*

$$\nabla^2\Psi_\rho(\Omega) = -\left[\nabla^2 F(\text{prox}_\Psi(\Omega)) + \rho W\right]^{-1}.$$

*In particular,* $\Psi_\rho$ *is* $\frac{1}{\mu}$*-smooth and* $\frac{1}{L+\rho\lambda_{\max}(W)}$ *strongly concave.*

The dual Moreau-envelope $\Phi_\rho$ and primal Moreau-envelope $\Psi_\rho$ are connected through the following relationship.

**Proposition 11.** *The gradient of the Moreau envelope* $\Phi_\rho$ *is given by*

$$\nabla\Phi_\rho(\mathbf{\Lambda}) = -\sqrt{\mathbf{W}}\nabla\Psi_\rho(\sqrt{\mathbf{W}}\mathbf{\Lambda}). \tag{14}$$

*Proof.* To simplify the presentation, let us denote

$$\mathbf{X}(\mathbf{\Lambda}) = \arg\min_{\mathbf{X}}\left\{F(\mathbf{X}) + (\sqrt{\mathbf{W}}\mathbf{\Lambda})^T\mathbf{X} + \frac{\rho}{2}\|\mathbf{X}\|_{\mathbf{W}}^2\right\} = \nabla\Psi_\rho(\sqrt{\mathbf{W}}\mathbf{\Lambda}).$$

From the optimality of $\mathbf{X}(\mathbf{\Lambda})$, we have

$$\nabla F(\mathbf{X}(\mathbf{\Lambda})) + \sqrt{\mathbf{W}}\mathbf{\Lambda} + \rho\mathbf{W}\mathbf{X}(\mathbf{\Lambda}) = 0$$

From the fact that $\nabla F(x) = y \Leftrightarrow \nabla F^*(y) = x$, we have

$$\mathbf{X}(\mathbf{\Lambda}) = \nabla F^*\left(-\sqrt{\mathbf{W}}\left[\mathbf{\Lambda} + \rho\sqrt{\mathbf{W}}\mathbf{X}(\mathbf{\Lambda})\right]\right).$$

Let $\mathbf{\Gamma} = \mathbf{\Lambda} + \rho\sqrt{\mathbf{W}}\mathbf{X}(\mathbf{\Lambda})$, then

$$-\sqrt{\mathbf{W}}\nabla F^*(-\sqrt{\mathbf{W}}\mathbf{\Gamma}) + \frac{1}{\rho}(\mathbf{\Gamma} - \mathbf{\Lambda}) = 0.$$

Therefore $\mathbf{\Gamma}$ is the minimizer of the function $F^*(-\sqrt{W}\Gamma) + \frac{1}{2\rho}\|\Gamma - \Lambda\|^2$, namely

$$\text{prox}_{\Phi_\rho}(\mathbf{\Lambda}) = \mathbf{\Lambda} + \rho\sqrt{\mathbf{W}}\mathbf{X}(\mathbf{\Lambda}).$$

Then based on the expression for the gradient in Prop 7, we obtain the desired equality (14). $\qquad\square$

Proposition 14 demonstrates that solving the augmented Lagrangian subproblem could be viewed as evaluating the gradient of the Moreau-envelope. Hence applying gradient descent on the Moreau-envelope gives the non-accelerated augmented Lagrangian framework Algorithm 1. Even more, applying Nesterov's accelerated gradient on the Moreau-envelope $\Phi_\rho$ yields accelerated Augmented Lagrangian Algorithm 4. In addition, when the subproblems are solved inexactly, this corresponds to an inexact evaluation on the gradient. This interpretation allows us to derive guarantees for the convergence rate of the dual variables. Before present the the convergence analysis in detail, we formally establish the connection between the primal solution and the dual solution.

**Lemma 12.** *Let* $x^*$ *be the optimum of* $f$ *and define* $\mathbf{X}^* = \mathbf{1}_n \otimes x^* \in \mathbb{R}^{nd}$. *Then there exists a unique* $\mathbf{\Lambda}^* \in Im(\mathbf{W})$ *such that* $\mathbf{\Lambda}^*$ *is the optimum of the dual problem* (D). *Moreover, it satisfies*

$$\nabla F(\mathbf{X}^*) = -\sqrt{\mathbf{W}}\mathbf{\Lambda}^*.$$

*Proof.* Since $Ker(W) = \mathbb{R}\mathbf{1}_n$, we have

$$Ker(\mathbf{W}) = Ker(W \otimes I_d) = Vect(\mathbf{1}_n \otimes \mathbf{e}_i, i = 1, \cdots, d),$$

where $\mathbf{e}_i$ is the canonical basis with all entries 0 except the $i$-th equals to 1. By optimality, $\nabla f(x^*) = \sum_{i=1}^{n} \nabla f_i(x^*) = 0$. This implies that $\nabla F(x^*)^T (\mathbf{1}_n \otimes \mathbf{e}_i) = 0$, for all $i = 1, \cdots d$. In other words, $\nabla F(X^*)$ is orthogonal to the null space of $\mathbf{W}$, namely $\nabla F(X^*) \in Im(\mathbf{W})$. Therefore, there exists $\mathbf{\Lambda}$ such that $\nabla F(\mathbf{X}^*) = -\mathbf{W}\mathbf{\Lambda}$. By setting $\mathbf{\Lambda}^* = \sqrt{\mathbf{W}}\mathbf{\Lambda}$, we have $\mathbf{\Lambda}^* \in Im(\mathbf{W})$ and $\nabla F(\mathbf{X}^*) = -\sqrt{\mathbf{W}}\mathbf{\Lambda}^*$. In particular, since $\nabla F(x) = y \Leftrightarrow \nabla F^*(y) = x$, we have,

$$\sqrt{\mathbf{W}}\nabla F^*(-\sqrt{\mathbf{W}}\mathbf{\Lambda}^*) = \sqrt{\mathbf{W}}\mathbf{X}^* = 0. \tag{15}$$

Hence $\mathbf{\Lambda}^*$ is the solution of the dual problem (D) and it is the unique one lies in the $Im(\mathbf{W})$. $\qquad\square$

Throughout the rest of the paper, we use $\Lambda^*$ to denote the unique solution as shown in the lemma above. We would like to emphasize that even though $F^*$ is strongly convex, the dual problem (D) is not strongly convex, because $W$ is singular. Hence, the solution of the dual problem is not unique unless we restrict to the image space of $\mathbf{W}$. To derive the linear convergence rate, we need to show that the dual variable always lies in this subspace where the Moreau-envelope $\Phi_\rho$ is strongly convex.

**Theorem 13.** *Consider the sequence of primal variables $(\mathbf{X}_k)_{k \in \mathbb{N}}$ generated by Algorithm 3 with the subproblem solved up to $\epsilon_k$ accuracy, i.e. Option I. Therefore,*

$$\|\mathbf{X}_{k+1} - \mathbf{X}^*\|^2 \leq 2\epsilon_{k+1} + C \left(1 - \sqrt{\frac{\mu_\rho}{L_\rho}}\right)^k \left(\sqrt{\mu_\rho \Delta_{dual}} + A_k\right)^2 \tag{16}$$

*where $\mathbf{X}^* = \mathbf{1}_n \otimes x^*$, $L_\rho = \frac{\lambda_{\max}(W)}{\mu + \rho\lambda_{\max}(W)}$, $\mu_\rho = \frac{\lambda_{\min}^+(W)}{L + \rho\lambda_{\min}^+(W)}$, $C = \frac{2\lambda_{\max}(W)}{\mu^2\mu_\rho^2}$, $\Delta_{dual}$ is the dual function gap defined by $\Delta_{dual} = F^*(-\sqrt{\mathbf{W}}\mathbf{\Lambda}_1) - F^*(-\sqrt{\mathbf{W}}\mathbf{\Lambda}^*)$ and $A_k = \sqrt{\lambda_{\max}(W)} \sum_{i=1}^{k} \sqrt{\epsilon_i} \left(1 - \sqrt{\frac{\mu_\rho}{L_\rho}}\right)^{-i/2}$.*

*Proof.* The proof builds on the concepts developed so far in this section. We start by showing that the dual variable $\mathbf{\Lambda}_k$ converges to the dual solution $\mathbf{\Lambda}^*$ in a linear rate. From the interpretation given in Prop 7 and Prop 11, the sequence $(\Lambda_k)_{k \in \mathbb{N}}$ given in Algorithm 2 is equivalent to applying Nesterov's accelerated gradient method on the Moreau-envelope $\Phi_\rho$. In the inexact variant, the inexactness on the solution directly translates to an inexact gradient of $\Phi_\rho$, where the inexactness is given by

$$\|e_k\| = \|\sqrt{\mathbf{W}}(X_k - X_k^*)\| \leq \sqrt{\lambda_{\max}(W)}\|X_k - X_k^*\| \leq \sqrt{\lambda_{\max}(W)\epsilon_k}.$$

Hence $(\Lambda_k)_{k \in \mathbb{N}}$ in Algorithm 4 is obtained by applying inexact accelerated gradient method on the Moreau-envelope $\Phi_\rho$. Note that by induction $\Lambda_k$ and $\Omega_k$ belong to the image space of $\sqrt{\mathbf{W}}$, in which the dual Moreau-envelope $\Phi_\rho$ is strongly convex. Following the analysis on inexact accelerated gradient method Prop 4 in [45], we have

$$\frac{\mu_\rho}{2}\|\mathbf{\Lambda}_{k+1} - \mathbf{\Lambda}^*\|^2 \leq \left(1 - \sqrt{\frac{\mu_\rho}{L_\rho}}\right)^{k+1} \left(\sqrt{2\Delta_{\Phi_\rho}} + \sqrt{\frac{2}{\mu_\rho}}A_k\right)^2 \tag{17}$$

where $\Delta_{\Phi_\rho} = \Phi_\rho(\Lambda_1) - \Phi_\rho^*$ and $A_k$ is the accumulation of the errors given by

$$A_k = \sum_{i=1}^{k} \|e_i\| \left(1 - \sqrt{\frac{\mu_\rho}{L_\rho}}\right)^{-i/2} \leq \sum_{i=1}^{k} \sqrt{\lambda_{\max}(W)\epsilon_i} \left(1 - \sqrt{\frac{\mu_\rho}{L_\rho}}\right)^{-i/2}.$$

Based on the convergence on the dual variable, we could now derive the convergence on the primal variable. Let $X_{k+1}^*$ be the exact solution of the problem $P_{k+1}$. Then

$$\begin{aligned}
\|\mathbf{X}_{k+1}^* - \mathbf{X}^*\| &= \|\nabla\Psi_\rho(\sqrt{\mathbf{W}}\mathbf{\Lambda}_{k+1}) - \nabla\Psi_\rho(\sqrt{\mathbf{W}}\mathbf{\Lambda}^*)\| \\
&\leq \frac{1}{\mu}\|\sqrt{\mathbf{W}}(\mathbf{\Lambda}_{k+1} - \mathbf{\Lambda}^*)\| \quad \text{(From Prop 10.3)} \\
&\leq \frac{\sqrt{\lambda_{\max}(W)}}{\mu}\|\mathbf{\Lambda}_{k+1} - \mathbf{\Lambda}^*\|. \tag{18}
\end{aligned}$$

Finally, from triangle inequality

$$\|\mathbf{X}_{k+1} - \mathbf{X}^*\|^2 \leq 2\|\mathbf{X}_{k+1} - \mathbf{X}_{k+1}^*\|^2 + 2\|\mathbf{X}_{k+1}^* - \mathbf{X}^*\|^2$$

$$\leq 2\epsilon_{k+1} + \frac{2\lambda_{\max}(W)}{\mu^2\mu_\rho}(1 - \sqrt{\kappa_\rho})^{k+1}\left(\sqrt{2\Delta_{\Phi_\rho}} + \sqrt{\frac{2}{\mu_\rho}}A_k\right)^2.$$

The desired inequality follows from reorganizing the constant and the fact that $\Delta_{\Phi_\rho} \leq \Delta_{dual}$. $\quad\square$

*Proof of Theorem 2.* Plugging in the choice of $\epsilon_k = \frac{\mu_\rho}{2\lambda_{\max}(W)}\left(1 - \frac{1}{2}\sqrt{\frac{\mu_\rho}{L_\rho}}\right)^k \Delta_{dual}$ in (16) yields the desired convergence rate.

$\square$

# D   Proof of Lemma 4

**Lemma 14.** *With the parameter choice as Theorem 2, then warm starting the $k$-th subproblem $P_k$ at the previous solution $\mathbf{X}_{k-1}$ gives an initial gap*

$$\|\mathbf{X}_{k-1} - \mathbf{X}_k^*\|^2 \leq 8\frac{C_\rho}{\mu_\rho}\epsilon_{k-1}.$$

*Proof.* From triangle inequality, we have

$$\|\mathbf{X}_{k-1} - \mathbf{X}_k^*\|^2 \leq 2(\|\mathbf{X}_{k-1} - \mathbf{X}^*\|^2 + \|\mathbf{X}_k^* - \mathbf{X}^*\|^2)$$

The desired inequality follows from the convergence on the primal iterates and (18), i.e.

$$\|\mathbf{X}_{k-1} - \mathbf{X}_k^*\|^2 \leq \frac{2C_\rho}{\mu_\rho}\epsilon_{k-1}, \quad \|\mathbf{X}_k^* - \mathbf{X}_k^*\|^2 \leq \frac{2C_\rho}{\mu_\rho}\epsilon_k.$$

$\square$

**Deriving the complexity in Table 2**   We have applied the same warm-start strategy in the complexity analysis of SSDA+AGD as IDEAL+AGD, the higher computation cost of SSDA is due to an intrinsic weakness of the method. The high level intuition is that the regularization parameter $\rho$ improves the condition number of the Moreau-envelope, which reduces the number of subproblems. More explicitly, the number of subproblems to be solved by IDEAL/SSDA is given by the formula

$$K = O\left(\sqrt{\frac{L_\rho}{\mu_\rho}}\log\left(\frac{C_\rho\Delta_{dual}}{\epsilon}\right)\right), \text{ where } L_\rho = \frac{\lambda_{\max}(W)}{\mu + \rho\lambda_{\max}(W)}, \mu_\rho = \frac{\lambda_{\min}^+(W)}{L + \rho\lambda_{\min}^+(W)}.$$

(Eq. 5 on page 5)

Ignoring the log factor, this quantity is proportional to the regularized condition number $\sqrt{\frac{L_\rho}{\mu_\rho}}$.

- For SSDA (which is equivalent to $\rho = 0$), we have $\sqrt{L_\rho/\mu_\rho} = \sqrt{\kappa_f\kappa_W}$;
- for IDEAL, by choosing $\rho = \frac{L}{\lambda_{\max}(W)}$, we have $\sqrt{L_\rho/\mu_\rho} \leq \sqrt{2\kappa_W}$.

Hence, IDEAL saves a factor of order $\sqrt{\kappa_f}$ compared to SSDA in the number of subproblems. Moreover, with the proposed choice of $\rho$, the cost of inexactly solving the subproblems is essentially the same for IDEAL and SSDA. Therefore we obtain the improvement in computation cost.

# E   Multi-stage algorithm: MIDEAL

Intuitively, we simply replace the mixing matrix $W$ by $Q(W)$, resulting in a better graph condition number. However, each evaluation of the new mixing matrix $Q(W)$ requires $deg(Q)$ rounds of communication, given by the AcceleratedGossip algorithm introduced in [43]. For completeness of the discussion, we recall this procedure in Algorithm 6. In particular, given $W$ and $X$, AcceleratedGossip($W$,$X$) returns $Q(W)X$, based on the communication oracle $W$.

**Algorithm 5** MIDEAL: Multi-stage Inexact Acc-Decentralized Augmented Lagrangian framework

**Input:** mixing matrix $W$, regularization parameter $\rho$, stepsize $\eta$, extrapolation parameters $\{\beta_k\}_{k\in\mathbb{N}}$

1: Initialize dual variables $\boldsymbol{\Lambda}_1 = \boldsymbol{\Omega}_1 = \mathbf{0} \in \mathbb{R}^{nd}$ and the polynomial $Q$ according to (7).
2: **for** $k = 1, 2, ..., K$ **do**
3:   $\mathbf{X}_k \approx \arg\min \left\{ P_k(\mathbf{X}) := F(\mathbf{X}) + \boldsymbol{\Omega}_k^T \mathbf{X} + \frac{\rho}{2}\|\mathbf{X}\|_{Q(\mathbf{W})}^2 \right\}$.
4:   $\boldsymbol{\Lambda}_{k+1} = \boldsymbol{\Omega}_k + \eta Q(\mathbf{W})\mathbf{X}_k$
5:   $\boldsymbol{\Omega}_{k+1} = \boldsymbol{\Lambda}_{k+1} + \beta_{k+1}(\boldsymbol{\Lambda}_{k+1} - \boldsymbol{\Lambda}_k)$
6: **end for**

**Output:** $\mathbf{X}_K$.

---

**Algorithm 6** AcceleratedGossip [43]

**Input:** mixing matrix $W$, vector or matrix $X$.

1: Set parameters $\kappa_W = \frac{\lambda_{\max}(W)}{\lambda_{\min}^+(W)}$, $c_2 = \frac{\kappa_W+1}{\kappa_W-1}$, $c_3 = \frac{2}{(\kappa_W+1)\lambda_{\min}^+(W)}$, # of iterations $J = \lfloor\sqrt{\kappa_W}\rfloor$.
2: Initialize coefficients $a_0 = 1$, $a_1 = c_2$, iterates $X_0 = X$, $X_1 = c_2(I - c_3 W)X$.
3: **for** $j = 1, 2, ..., J-1$ **do**
4:   $a_{j+1} = 2c_2 a_j - a_{j-1}$
5:   $X_{j+1} = 2c_2(1 - c_3 W)X_j - X_{j-1}$
6: **end for**

**Output:** $X_0 - \frac{X_J}{a_J}$.

---

# F   Implementation of Algorithms

We include in the following (Algorithm 7) an implementable version of the IDEAL+AGD algorithm to facilitate re-implementation of our framework.

---

**Algorithm 7** Implementation: IDEAL+AGD solver

**Input:** number of iterations $K > 0$, gossip matrix $W \in \mathbb{R}^{n\times n}$

1: $\omega_i(0) = \vec{0}$, $\gamma_i(0) = \vec{0}$, $x_i(0) = \overline{x_i}(0) = x_0$ for any $i \in [1, n]$
2: $\kappa_{inner} = \frac{L+\rho\lambda_{\max}(W)}{\mu}$, $\beta_{inner} = \frac{\sqrt{\kappa_{inner}}-1}{\sqrt{\kappa_{inner}}+1}$, $\kappa_\rho = \frac{L+\rho\lambda_{\min}^+(W)}{\mu+\rho\lambda_{\max}(W)}\frac{\lambda_{\max}(W)}{\lambda_{\min}^+(W)}$, $\beta_{outer} = \frac{\sqrt{\kappa_{outer}}-1}{\sqrt{\kappa_{outer}}+1}$
3: **for** $k = 1, 2, ..., K$ **do**
4:     Inner iteration: Approximately solve the augmented Lagrangian multiplier.
5:       Set $x_{i,k}(0) = y_{i,k}(0) = x_i(k-1)$, $\overline{x_{i,k}}(0) = \overline{y_{i,k}}(0) = \sum_{j\sim i} W_{ij}x_{j,k}(0)$
6:       **for** $t = 0, 1, ..., T-1$ **do**
7:         $x_{i,k}(t+1) = y_{i,k}(t) - \eta(\gamma_i(k) + \nabla f_i(y_{i,k}(t)) + \rho\overline{y_{i,k}}(t))$
8:         $y_{i,k}(t+1) = x_{i,k}(t+1) + \beta_{inner}(x_{i,k}(t+1) - x_{i,k}(t))$
9:         $\overline{y_{i,k}}(t+1) = \sum_{j\sim i} W_{ij}y_{j,k}(t+1)$
10:      **end for**
11:      Set $x_i(k) = x_{i,k}(T)$, $\overline{x_i}(k) = \sum_{j\sim i} W_{ij}x_{j,k}(T)$
12:    Outer iteration: Update the dual variables on each node
13:      $\lambda_i(k+1) = \omega_i(k) + \rho\overline{x_i}(k)$
14:      $\omega_i(k+1) = \lambda_i(k+1) + \beta_{outer}(\lambda_i(k+1) - \lambda_i(k))$
15: **end for**

**Output:**

---

# G   Further Experimental Results

Figure 3: Network Structures: **Left:**Circular graph with 4 nodes. **Right:**Barbell graph with 8 nodes.

Figure 4: Ablation study on the **regularization parameter** $\rho$ in IDEAL framework. For all the experiments, we use AGD as inner loop solver and set the same parameters as predicted by theory. We observe that when $\rho$ is selected in the range $[0.5\rho_{\mathrm{default}}, 10\rho_{\mathrm{default}}]$, the perfomance of the algorithm is quite similar and robust. We also observe that using a small $\rho$ degrades the performance of the algorithm, this phenomenon is consistent with the observation that the inexact SSDA [43] does not perform well since it uses $\rho = 0$. Another observation is that with larger $\rho$, such as $\rho = 2\rho_{\mathrm{default}}$ or $10\rho_{\mathrm{default}}$, the algorithm is more stable with less zigzag oscillation, which is preferable in practice.

Figure 5: Ablation study on the **inner loop complexity** $T_k$ in IDEAL framework. When the inner loop iteration is small, the algorithm becomes less stable, so we have decreased the momentum parameters to ensure the convergence. For these experiments, we use AGD solver with $\beta_{in} = 0.8$ and $\beta_{out} = 0.4$. As we can see, it is beneficial to perform multiple iterations in the inner loop rather than taking $T$=1 as in the EXTRA algorithm [47].

Figure 6: Test accuracy on MNIST task.