[Reviews · NeurIPS 2020]

Review 1

Summary and Contributions: This paper proposed a novel inexact augmented Lagrangian framework for decentralized convex optimziation, with accelerated gradient descent innerloops to approximately solve the subproblems, and an extra outer-loop momentum step for acceleration. A theoretical analysis is provided showing worst-case optimal and state-of-the-art convergence results for decentralized convex problems. Numerical results in this paper demonstrate and confirm such computational and communication benefits suggested by the theory.

Strengths: The paper is well-written with very solid theoretical contributions which will be of interest for the community. Compared to previous approaches for decentralized optimziation, the proposed method adopts double-loop acceleration (that is, inexact accelerated proximal point outerloops plus accelerated gradient inner-loops), which admits improvement over state-of-the-art methods both in theory and in practice.

Weaknesses: There is some room for improvement in presenting the numerical experiments. The numerical results only include the objective gap convergence curves, while the reviewer believes that it is better to also include the test error results to fully visualize the practical value for the proposed scheme. On the discussion of the stochastic inner-loop solver, the authors only considered the use of vanilla SGD which only converges sublinearly. However it is now well-known that with variance-reduction techniques such a convergence rate can be linear and have squared dependent on condition number. Hence the reviewer believes that the results in Table 1 3rd column is suboptimal and can be substantially improved.

Correctness: The proof seems correct after a quick check.

Clarity: Yes, the presentation of the paper is clear and easy to follow.

Relation to Prior Work: Yes, the discussion w.r.t. previous state-of-the-art decentralized algorithms is clearly provided.

Reproducibility: Yes

Additional Feedback:


Review 2

Summary and Contributions: This paper introduces a framework for designing primal distributed methods by the accelerated augmented Lagrangian method. When coupled with accelerated gradient descent in the inner loop, optimal convergence rates can be obtained (up to some log constant)

Strengths: 1. The theory is solid and the rate is the tightest to my knowledge among the papers appearing online before the submission deadline of NIPS. 2. The result is significant. 3. It is relavant to the NIPS community.

Weaknesses: I have two major concerns: 1. Similar to EXTRA, the inner loop of IDEAL also needs the communications between different nodes. So the communication cost and the computation cost of IDEAL+AGD should be the same in Table 2. In other words, the communication cost also hides the log factor. As a comparison, SSDA does not need communications in the inner loop. Thus, I wonder whether the communication cost is lower than that of SSDA in practice. I suggest the authors to plot the communication cost and computation cost separately. 2. I wonder whether SSDA+AGD+warm start can achieve the same computation cost by using the proof technique proposed in this paper. If it is, please clarify that the higher computation cost of SSDA is just beacuse the weakness of the proof, rather than the method itselt. If it is not, please give some details in the supplementary material, for example, the analysis of the computation cost of SSDA+AGD+warm start. It might be better to explain why IDEAL+AGD+warm start is theoretically better than SSDA+AGD+warm start (I guess it is because of rho=0 in SSDA?). Then, it may support the superioty of the proposed method. Minor comments: 1. line 113. The best rate for non-accelerated method is O( (kappa_f + kappa_w)log(1/eps) ). Xu, J., Tian, Y., Sun, Y., and Scutari, G. (2020). Distributed algorithms for composite optimization: Unified and tight convergence analysis. arXiv preprint arXiv:2002.11534. Li, H. and Lin, Z. Revisiting extra for smooth distributed optimization. SIAM Journal on Optimization, 30(3):1795-1821, 2020. arXiv:2002.10110. 2. The citation on line 506 may be incorrect. 3. Please give more details for the analysis of IDEAL+AGD, for example, why choose rho=L/lambda_max. 4. It might be better to give the rate for IDEAM+SVRG 5. It might be better to give the explicit log cost in Table 2, even if it is a constant of O( log (kappa_f kappa_W) ).

Correctness: Correct

Clarity: Yes

Relation to Prior Work: Yes

Reproducibility: Yes

Additional Feedback: After rebuttal: Thanks for the response. It has addressed my questions.


Review 3

Summary and Contributions: The authors consider a decentralized optimization problem in which the loss function is a sum of individual loss functions, each corresponding to a node in a graph. The loss functions must be smooth, strongly convex, and have gradients which are "L-Lipschitz", and connected nodes can pass information to each other during the optimization (it is assumed that the graph is given, i.e., fixed). The authors derive a "Inexact Accelerated Decentralized Augmented Lagrangian framework" (IDEAL) which achieves recently derived lower bounds (in terms of condition number of the objective function and the condition number of the communication matrix) on computational complexity to achieve an epsilon-exact approximation to the unique solution of this problem. The method is primal and does not require solving the dual problem. They provide an analysis and proof of the computational complexity, and compare to existing methods which they demonstrate are specific examples of IDEAL.

Strengths: This work appears to provide the first numerical method that achieves the theoretical lower bound on the computational complexity. The theoretical proof of this lower bound is fairly influential, so I believe that the proposed algorithm is an important milestone. Full proofs and convergence analysis are given. Existing methods -- SSDA and MSDA -- are shown to special cases, unifying the analysis of several methods under the one given in this work. Numerical examples for different parameters are given for an MNIST benchmark which are consistent with the analysis of the proposed algorithm.

Weaknesses: The contribution of this article is strong, but my main issue is I don't believe this is the best format for presenting these results. The authors have done a good job moving many technical details and proofs into the appendices, but there are many technical statements and discussions in the main text which are not sufficiently fleshed out and difficult to verify or follow, probably due to lack of space. An example of this is Table 2. The results in this table are important for comparing the proposed method with existing methods, a main point of the article. But after reading this article, as someone who is not a dedicatged expert on this area, it would be very difficult for me to derive and verify the formulas in this table. The "Experiments" section also suffers from this reason -- although the trends in the results are adequetly explained, the main text of the article is not sufficient to understand the setup for these experiments. The supplemental material doesn't solve this issue in my opinion; there is a mention of a convolutional kernel network and a reference is given, but I believe there should be a reasonably self-contained description of the experiment and model in the supplement. It is true that the authors provide full code, but it shouldn't be necessary to dig through the code just to understand the setup. It may be possible to solve these issues by expanding the supplemental material to explain all the technical details and experiment in a self-contained way.

Correctness: The analysis appears correct and is consistent with the numerical experiments.

Clarity: Some technical details (see above) are not fully fleshed out, but the rest of the paper is fairly clear. Some minor issues: The definition of "L-Lipschitz" is not clearly stated. What is the Omega in Theorem 1?

Relation to Prior Work: The authors do a good job framing their contribution in relation to previous work, and a thorough review is provided.

Reproducibility: Yes

Additional Feedback:


Review 4

Summary and Contributions: In this work, the authors propose a decentralized optimization framework that achieves optimal convergence rate using the primal formulation only. The framework also allows the sub-problem to be solved approximately. To achieve optimal computation cost, the authors further propose a multi-stage variant of the proposed framework. Connections to existing work are discussed in details. In particular, comparisons with an inexact version of SSDA/MSDA are also discussed. In experiments, different communication cost regimes are considered and results validate the theoretical findings.

Strengths: A unified framework is proposed for decentralized optimization based on the accelerated augmented Lagrangian method. Optimal rates are achieved both in terms of communication cost and computation cost. The connections with existing work are discussed in details.

Weaknesses: Experiments are only conducted on one task. More tasks with different data dimensions and condition numbers could be more compelling and helpful to understand how different algorithms perform in practice.

Correctness: I did not check the proof in details, but the outline of the theoretical analysis make sense to me.

Clarity: The paper is fairly well-written, with connections to existing method clearly discussed. The necessary technical details are also included such that the paper is accessible to audience who does not go into the proof details.

Relation to Prior Work: Yes.

Reproducibility: Yes

Additional Feedback: ----------------After author response------------ The response addresses my concerns on experiments. Hence I am raising my score from 6 to 7.

[Author Response · NeurIPS 2020]

We appreciate the positive feedbacks from all the reviewers and provide a detailed response as follows.

**R1: "Room for improvement in presenting numerical experiments ... it is better to include test error results"**

Our initial intention was to conduct experiments reflecting the convergence rate in the theoretical analysis. We
appreciate your suggestions, and will include the test error in the supplementary material.

**R1/R2: "It would be interesting to include results using a variance reduction technique"**

We thank both reviewers for mentioning variance reduction techniques, this is indeed very related. To apply
variance reduction methods, we need to further assume that the loss function on each node can be decomposed
into a finite-sum structure, say each loss is a sum of $m$ functions. In this case, combining IDEAL+SVRG will
provide a complexity of order $\tilde{O}\left((m + \kappa_f)\sqrt{\kappa_W}\log(1/\epsilon)\right)$, where acceleration is achieved with respect to the
graph condition number $\kappa_W$ but not with respect to objective's condition number $\kappa_f$ (see also 'Two-fold acceleration'
on page 6). It is conceivable that there might exist a better design for finite sum structure, e.g., by communicating
the full gradient evaluated in SVRG, etc. This is definitely an interesting direction which we leave to future work.

**R2: "I wonder whether the communication cost is lower than that of SSDA in practice. I suggest the authors**
**to plot the communication cost and computation cost separately. "**

When the computation/communication $\tau$ is large, the dominant term is the communication cost. Hence the third
column ($\tau = 10$) in Figure 1 roughly reflects the communication cost. Thanks for the suggestion, we will include a
separate plot in the supplementary material.

**R2: "I wonder whether SSDA+AGD+warm start can achieve the same computation cost by using the proof**
**technique proposed in this paper. "**

We appreciate the reviewer raising this question, this is indeed one of the core messages we would like to convey:
IDEAL improves upon SSDA in a non-trivial way. In short, we have applied the same warm-start strategy as
IDEAL+AGD in the complexity analysis of SSDA+AGD, the higher computation cost of SSDA is due to an
intrinsic weakness of the method. The high level intuition is that the regularization parameter $\rho$ improves the
condition number of the Moreau-envelope, which reduces the number of subproblems. More explicitly, the number
of subproblems to be solved by IDEAL/SSDA is given by the formula

$$K = O\left(\sqrt{\frac{L_\rho}{\mu_\rho}}\log\left(\frac{C_\rho \Delta_{dual}}{\epsilon}\right)\right), \text{ where } L_\rho = \frac{\lambda_{\max}(W)}{\mu + \rho\lambda_{\max}(W)}, \mu_\rho = \frac{\lambda_{\min}^+(W)}{L + \rho\lambda_{\min}^+(W)}. \qquad \text{(Eq. 5 on page 5)}$$

Ignoring the log factor, this quantity is proportional to the regularized condition number $\sqrt{\frac{L_\rho}{\mu_\rho}}$.

- For SSDA (which is equivalent to $\rho = 0$), we have $\sqrt{L_\rho/\mu_\rho} = \sqrt{\kappa_f \kappa_W}$;
- for IDEAL, by choosing $\rho = \frac{L}{\lambda_{\max}(W)}$, we have $\sqrt{L_\rho/\mu_\rho} \leq \sqrt{2\kappa_W}$.

Hence, IDEAL saves a factor of order $\sqrt{\kappa_f}$ compared to SSDA in the number of subproblems. Moreover, with the
proposed choice of $\rho$, the cost of inexactly solving the subproblems is essentially the same for IDEAL and SSDA.
Therefore we obtain the improvement in computation cost.

**R2: "How to determine the regularization parameter, why choose $\rho = L/\lambda_{max}$"**

The global complexity of IDEAL is given by the product of $K$ (number of subproblems) and $T$ (number of iterations
for each subproblem), both of them are functions of $\rho$. The way we determine the regularization parameter is to
minimize the global complexity with respect to the parameter $\rho$, then simplify it in an asymptotic manner.

**R2: "References"**

Thanks for pointing out these references, we will include them in the revision and some other recent works as well.

**R3: "The contribution of this article is strong, but my main issue is I don't believe this is the best format for**
**presenting these results... it would be very difficult for me to derive and verify the formulas in this table 2"**

We are grateful to the reviewer's suggestion and will provide a more detailed explanation of this derivation in the
revision. Please kindly check our response to R2's question starting in line 18 for a high level justification.

**R3:" The main text of the article is not sufficient to understand the setup for these experiments"**

All the experiments we conducted are for binary logistic regression. For the MNIST experiment, we directly use the
normalized image as input feature. For the CIFAR experiment, the dataset is much more complicated, so a linear
model is not rich enough. Hence, we first apply an unsupervised learning model (convolutional kernel network) to
extract the feature and then apply the logistic regression on top of it. This can be approximately viewed as training
the last layer of a conventional neural network by freezing the (well-trained) first layers. We will improve our
presentation on experimental settings with more details in the revision.

**R4: "Experiments are only conducted on one task. More tasks could be more compelling..."**

Due to limited space, we have only presented the MNIST experiment in the main paper, however the CIFAR
experiment and other ablation studies can be found in the supplementary material.

[Meta-Review · NeurIPS 2020]

Good paper, all reviewers are convinced.